



# A new satellite-derived dataset for marine aquaculture areas in the China's coastal region

Yongyong Fu[1], Jinsong Deng[1], Hongquan Wang[1], Alexis Comber[2], Wu Yang[1], Wenqiang Wu[1],Shixue You[1], Yi Lin[3], Ke Wang[1]

[1]College of Environmental and Resource Sciences, Zhejiang University, Hangzhou, 310058, China
[2]School of Geography, University of Leeds, Leeds LS1 9JT, UK
[3]Department of Geography, University of Hong Kong, Hong Kong SAR 999077, China

*Correspondence to*: Jinsong Deng (jsong_deng@zju.edu.cn)

**Abstract.** China has witnessed extensive development of the marine aquaculture industry over recent years. However, such
rapid and disordered expansion posed risks to coastal environment, economic development, and biodiversity protection. This
study aimed to produce an accurate national-scale marine aquaculture map at a spatial resolution of 16 m, using a proposed
deep convolution neural networks (CNNs) based model and applied it to satellite data from China's GF-1 sensor in an end-
to-end way. The analyses used homogeneous CNNs to extract high-dimensional features from the input imagery and
preserve information at full resolution. Then, a hierarchical cascade architecture was followed to capture multi-scale features
and contextual information. This hierarchical cascade homogeneous neural network (HCHNet) was found to achieve better
classification performance than current state-of-the-art models (FCN-32s, Deeplab V2, U-Net, and HCNet). The resulting
marine aquaculture area map has an overall classification accuracy >95% (95.2%-96.4, 95% confidence interval). And
marine aquaculture was found to cover a total area of ~1100 km$^2$ (1096.8 km$^2$-1110.6 km$^2$, 95% confidence interval) in
China, of which more than 85% are marine plant culture areas, with 87% found in the Fujian, Shandong, Liaoning, and
Jiangsu provinces. The results confirm the applicability and effectiveness of HCHNet when applied to GF-1 data, identifying
notable spatial distributions of different marine aquaculture areas and supporting the sustainable management and ecological
assessments of coastal resources at a national scale. The national-scale marine aquaculture map at 16 m spatial resolution is
published in the Google Maps kmz File Format with georeferencing information at https://doi.org/10.5281/zenodo.3881612
(Fu et al., 2020).

## 25    1 Introduction

Marine aquaculture, which refers to the breeding, rearing, and harvesting of aquatic plants or animals in marine waters, has
significant potential for food production, economic development, and environmental protection in coastal areas (Burbridge et
al., 2001; Campbell and Pauly, 2013; Gentry et al., 2017). It has become a fast-growing industry in China due to the
significant increase in the demand for seafood, support from polices, and technology innovation (Liang et al., 2018). The
marine aquaculture production in China has increased from 10.6 million tons in 2000 (Bureau of Fisheries of the Ministry of



Agriculture, 2001) to 20.0 million tons in 2017 (Bureau of Fisheries of the Ministry of Agriculture, 2018). However, such rapid and disordered growth may cause severe economic losses and environmental problems, such as water pollution (Tovar et al., 2000), biodiversity decrease (Galil, 2009; Rigos and Katharios, 2010), and marine sediment pollution (Porrello et al., 2005; Rubio-Portillo et al., 2019). Therefore, accurate mapping and monitoring of marine aquaculture can provide evidence
to support the sustainable management of coastal marine resources.

Previous research in this domain can be grouped into visual interpretation, analyses enhanced by including ancillary data such as information about spatial structure, object-based image analysis (OBIA), and deep learning based methods. Visual interpretation is used less frequently as it requires too much time and effort. Enhanced analyses that incorporate features such as texture or average filtering (Fan et al., 2015; Lu et al., 2015; Xiao et al., 2013) are commonly employed for pixel-
based approaches. However, these are subject to noise (the salt-and-pepper effect) and decreased accuracy (Zheng et al., 2017). OBIA has been widely used for the detailed interpretation of marine aquaculture from remote sensing images (Fu et al., 2019a; Wang et al., 2017; Zheng et al., 2017). It first partitions the image into segments and then classifies segments based on their internal properties (Blaschke et al., 2014). However, since almost all these methods are proposed based on the handcrafted features, it is inherently difficult for them to achieve balance between high discriminability and good robustness
(L. Zhang et al., 2016). To solve such problems, the remote sensing community has started to incorporate deep fully convolutional neural networks (FCN) within marine aquaculture detection tasks using high spatial resolution (HSR) images at local scales (Cui et al., 2019; Fu et al., 2019c; Shi et al., 2018). However, the opportunities associated with analyses of the high volumes of publicly available and free remote sensing data at medium resolution, such as Landsat, Sentinel-2 A/B, and GaoFen-1 wide-field-of-view (GF-1 WFV) imagery, have not been exploited. Therefore, it is necessary to develop a
detection system applying deep FCNs to such data to provide more reliable and effective mapping and monitoring over wider areas, supporting evaluations of marine aquaculture areas at a national scale.

However, there are several critical limitations for accurate mapping of marine aquaculture areas using deep FCN-based methods when applied to medium-resolution data. The first is the coexistence of multi-scale objects, such as the large sea areas as well as small aquaculture areas, making it difficult to focus FCN on small marine aquaculture objects. A common
approach is to use inputs of different sizes from the original images (Eigen and Fergus, 2015; Liu et al., 2016; Zhao and Du, 2016) to ensure that different object sizes are prominent in different parts of the FCN structure, but such methods take more time due to the repetitive sampling of the input imagery. Some researchers have generated multi-scale features using atrous convolution (Chen et al., 2018) or pooling operations at different scales (He et al., 2015; Zhao et al., 2017). However, such approaches may be limited to a certain range of receptive fields, as operations may be applied to invalid zones when pooling
with a larger pooling size or atrous convolution with a higher atrous rate. The second critical limitation is that the final features may have a smaller size than the input imagery due to consecutive pooling operations in FCN, making it hard to identify land cover details. To solve this problem, researchers have used deconvolution operation (Noh et al., 2015) or fused features (Pinheiro et al., 2016; Ronneberger et al., 2015), but FCN may fail to identify relatively small marine aquaculture areas. Finally, some researchers have tried to refine the classification approach by including known boundaries (Bertasius et



al., 2016; Fu et al., 2019b; Marmanis et al., 2018), but such methods require additional classification steps to perform boundary extraction.

In conclusion, although present methods have been successfully applied for dense classifications, the challenge of using them to accurately extract the marine aquaculture areas from medium resolution images at a national scale remains. Some of the problems associated with this technique and domain can be solved by incorporating multi-scale contextual information while avoiding shrinking of the target features. Inspired by this idea, this study has developed a new method, the hierarchical cascade homogeneous neural network (HCHNet) model, for the detection of marine aquaculture areas at national scale from medium resolution imagery. The rest of the paper proceeds as follows. Section 2 briefly presents a description of the study area and different types of marine aquaculture. Section 3 introduces the input data and method to develop the proposed deep learning architectures, implementation details, and methodological choices. The results are presented in Section 4. Section 5 then discusses the methods and the limitations of using deep learning methods with medium resolution data, and finally, Section 6 concludes the paper.

## 2 Study area

The study area included all of the potential marine aquaculture areas in China's coastal regions (Fig. 1). Due to the large amount of coastline and associated resources, many coastal marine aquaculture areas have rapidly developed in coastal regions. After a visual inspection on the HSR images from Google Earth, we empirically set the width of the study area along with the coastal line for detection as 30 km. According to the types of cultivated aquatic products, the marine aquaculture areas in China can be classified into marine animal culture (MAC) areas and marine plant culture (MPC) areas. MAC areas are cultured with marine animals, such as fish, crustaceans, shellfish, etc., in connected cages (Fig. 2k), or wooden rafts (Fig. 2d). Most of the cages and rafts are small (normally 3 × 3 or 5 × 5 m in size) and simple in form (normally square). The materials used to construct these cages are collected locally and include bamboo, wooden boards, plastic foam floats, and polyvinyl chloride or nylon nets. Because of the low investment costs and ease of construction, farmers typically make the cages themselves. As they cannot withstand waves generated by typhoons or sea currents, most of cages must be installed in inshore waters and sheltered sites (Fig. 2c, i).

MPC areas are generally cultivated with seaweed, such as kelp, undaria, gracilaria, etc. Most of the seaweed is twisted around ropes about 2 m in length. The ropes are linked or tied to one (Fig. 2 f) or two floating lines (Fig. 2j), which are about 60 m long and kept at the sea surface by buoys made from foam or plastic and anchored by lines tied to wooden pegs driven into the sea bottom. As most of the MPC are submerged in the sea water, the features of MPC in remotely sensed images are usually influenced by different environment (Fig. 2b, e, g, h , j), making it difficult for classification.



## 3 Materials and methods

Due to the large number of factors that could potentially affect classification performance, implementation of the FCN-based method at the national scale is a challenge. To reduce the influence of various land covers, we used the coastal line vector (Chuang et al., 2019) to exclude mainland areas after preprocessing all the input images (Fig. 3); then, we produced the marine aquaculture map by utilizing the HCHNet method, which was trained and tested on dataset validated by field survey or HSR images.

### 3.1 Data and preprocessing

In this study, images from the WFV sensors of GF-1 were selected as the primary data source. This satellite carries four integrated WFV sensors, providing multi-spectrum data with a two-day revisit cycle and a swath width of 800 km when the four sensors are combined. Each WFV sensor has four multi-spectral bands at 16 m spatial resolution: B1 (450−520 nm, blue), B2 (520−590 nm, green), B3 (630−690 nm, red), and B4 (770−890 nm, near infrared). A total of 35 quantified GF-1
WFV images spanning the 2016−2019 period were finally selected from the China Centre for Resources Satellite Data and Application to cover the whole coastal region in China and to filter for cloud coverage (Fig. 1). The product filenames are listed in Table S1 of the Supplementary material.

The images were projected into the UTM map projection, and atmospheric correction was undertaken using the FLAASH atmospheric correction model embedded in the ENVI software (v5.3.1). A 30 km buffer was used to extract the images of
the coastal areas, and the final set of clipped images consisted of four image bands at a spatial resolution of 16 m, used as inputs in the following parts.

### 3.2 Hierarchical cascade homogeneous neural network

As shown in Fig. 4, the proposed HCHNet is an FCN-based network, which can be trained and applied to a large area in an end-to-end way. Specifically, a homogeneous CNN was designed to extract high-dimensional features from the input images.
A hierarchical cascade structure was followed to extract multi-scale contextual information gradually based on high-dimensional features. The following subsections introduce three important components of the proposed HCHNet method, including (1) an encoder based on a homogeneous CNN; (2) hierarchical cascade structure; and (3) a loss function.

### 3.2.1 Encoder based on a homogeneous CNN

Traditional CNN uses down-sampling process to improve the local invariance, and the prediction results are usually only
labels at the patch level. For semantic segmentation, FCN can enlarge the down-sampling feature maps to full-sized outputs by using interpolation (Badrinarayanan et al., 2017) or deconvolution (Noh et al., 2015). However, foreground objects, such as the marine aquaculture areas in our study, occupy a smaller portion of the GF-1 WFV images than in the natural images, making it hard for the FCN to recover the details missing from consecutive pooling operations via learning.

As representations with high-resolution are important for the preservation of detailed information, the homogeneous CNN
(Shi et al., 2018) was used as the encoder. One of the advantages of the homogeneous CNN is that it retains the full
resolution of features by removing all of the pooling operations. As shown in Fig. 4, we built the encoder based on the
widely used VGG-16 model. The VGG-16 is constructed of thirteen convolutional layers and followed by three fully
connected layers. To preserve the spatial information and control the model size, the fully connected layer was removed and
convolutional kernels in the corresponding layers was reduced. As a result, the encoder can preserve full resolution features
as the input image.

### 3.2.2 Hierarchical cascade structure

Although removing pooling operations can preserve more detailed information, it can also decrease the receptive field of the
underlying neural network (Liu et al., 2018). In this case, with fixed and limited receptive fields, it may cause more
misclassifications because of the loss of multi-scale contextual information. To solve this problem, the hierarchical cascade
structure proposed in a previous study (Fu et al., 2019c) was used. This structure generally enlarges the receptive field and
increases the sampling rate by creating a hierarchical cascade structure using the atrous convolution layers (as shown in the
central part of Fig. 4). To reduce memory usage, batch normalization operations were used to replace the attention modules,
allowing feature maps from different levels to be concatenated and easing the training process (Ioffe and Szegedy, 2015).
Each atrous convolution layer in the proposed structure is formulated as follow:

$$F_1 = C_{k,D_1}[F_o], \tag{1}$$

$$F_l = C_{k,D_l}[\mathcal{L}(F_o \, \textcircled{C} \, F_1 \, \textcircled{C} \, F_2 \, \textcircled{C} \, F_3 \, \textcircled{C} \, ... \, \textcircled{C} \, F_{l-1})], l > 1, \tag{2}$$

$$D_1 < D_2 < D_3 < ... < D_l, \tag{3}$$

where $F_o$ denotes the feature maps from the output of our encoder network. $C_{k,D_l}[\cdot]$ denotes an atrous convolution operation
with the kernel size of $K \times K$ and dilation rate of $d$ at the $l$-level. $F_l$ ($l = 1,…$ , n) denotes the features at the $l$-level in the
structure. ' $\textcircled{C}$ ' denotes the concatenation operation. '$\mathcal{L}(\cdot)$' denotes the batch normalization. $D_l$ denotes the dilation rate
value at the $l$-level.

### 3.2.3 Loss function

A significant problem during the training of FCN is the imbalance of classes. Such imbalances can make training inefficient,
with relatively small marine aquaculture areas contributing little to the model training process. In contrast, much of the
coastal area contains negative samples, such as the sea area, which may dominate the training process and decrease marine
aquaculture identification accuracy. To address this class imbalance problem, Eigen and Fergus (2015) proposed reweighting
each class based on a loss calculation. Following this idea, the weighted loss was used to deal with the class imbalance
problem and to allow effective training of all examples. The loss was defined as:



$$WL(p,q) = -\sum_{i=1}^{n} \alpha_i p(x_i) \log(q(x_i)), \qquad (4)$$

where $p \in \{0,1\}$ represents the ground truth class, $q \in [0,1]$ is the predictive probability from the model for the classes, and $\alpha_i$ represents the weight for each class.

### 3.3 Implementation details

As shown in Fig. 4, the encoder was first applied, which is a homogenous CNN with 13 layers (Table 1), to produce high-dimensional and abstract features with full resolution from the input imagery. And then, multi-scale contextual information was captured by using the hierarchical cascade structure with atrous rates of 3, 6, and 9. To regulate the model's memory consumption and to prevent it growth too wide, $1 \times 1$ kernels were employed in the hierarchical cascade structure to keep all the channels of concatenated features fixed to 128, which have the same output feature maps of other atrous convolution layers.

As for the training and testing of HCHNet, we selected a total of 705 patches. Each one of them has non-overlapping 256 × 256 pixels from raw images (Fig. 1). The ground truth maps for each patch were obtained by visual interpretation. From them, we randomly selected 80% to construct the training dataset. Considering the relatively small training dataset, data augmentation was applied to make the training process more effective and reduce overfitting: each patch was flipped in the horizontal and vertical direction and was rotated counterclockwise by 90. As a result, there were 4656 patches can be used for the training of HCHNet.

In the experiment, we trained the HCHNet for 30 epochs using a batch size of 4 and the Adam optimizer. The Adam parameters were set as: β1 = 0.9, β2 = 0.999, and a learning rate of 0.0001. The HCHNet was built and implemented using the Keras (v2.2.4) on top of Tensorflow (v1.8.0). All of the experiments were undertaken on a computer with an graphics processing unit (GPU) of NVIDIA GeForce GTX 1070.

### 3.4 Comparing methods and accuracy assessment

#### 3.4.1 Comparing methods

To assess the effectiveness and advantage of our proposed method, we provided a comparison with four state-of-the-art FCN-based methods. We summarized the main information as follows:

FCN-32s: the first FCN-based method proposed by Zhang et al. (2015) for semantic segmentation. It was constructed based on the VGG-16, in which the original fully connected layers are convolutionized. The model predicts classification results by upsampling the final feature maps 32 times directly. It does not use any structure to extract multi-scale feature maps or get more detailed information from the shallow layers. Thus, it can be used as a baseline model for our proposed HCHNet methods.



U-Net: a typical FCN-based model with encoder−decoder structure,which was proposed by Ronneberger et al. (2015) for
semantic segmentation of medical images. The encoder has a similar structure to VGG-16. Different from the FCN 32s, U-
Net combined the feature maps in the decoder and mirrored feature maps in the encoder by using long-span connections to
provide precise localization and high classification accuracy.

Deeplab V2: Chen et al. (2018) proposed the Deeplab V2 (VGG-16 as the backbone) model for semantic segmentation,
which used the atrous spatial pyramid pooling (ASPP) structure to capture multi-scale contextual information, and then used
the fully connected conditional random fields (CRFs) as a post-processing tool to refine the prediction results.

HCNet: HCNet was proposed by Fu et al. (2019c) to map the detailed spatial distribution of marine aquaculture from HSR
images. This model has a variant of VGG-16 as an encoder, in which the stride and padding of the last two pooling layers are
set as one for high- resolution feature maps. It combines the long-span connections, while also combining a hierarchical
cascade structure.

The above models are suitable for classification and comparison purposes, because nearly all of these methods are VGG-16-
or similar structure-based neural networks. In the training phase, all of the above models were trained from scratch using the
same patches and experimental settings as in the HCHNet method.

### 3.4.2 Accuracy assessment and comparison

To perform the accuracy assessment, we followed the widely used random stratified sampling method (Padilla et al., 2014;
Ramezan et al., 2019) to generated 4000 randomly selected points in the coastal zone. Based on the visual interpretation
results from HSR images, several commonly-used accuracy statistics were calculated from the error matrix, such as producer
accuracy (PA), user accuracy (UA), overall accuracy (OA), and the kappa coefficient. Meanwhile, we also calculated the
area accuracy based on the test dataset, which accounts for 20% of the total samples.

After that, we compared the performances of our proposed method with four state-of-the-art FCN-based models. The
accuracy comparison was undertaken using the test dataset. To provide a quantitative assessment between our proposed
method and other methods, we calculated the widely-used F1 score (F1), precision, and recall (POWERS, 2011) as follows:

$$\text{Precision} = \frac{\text{TP}}{\text{TP} + \text{FP}}. \tag{9}$$

$$\text{Recall} = \frac{\text{TP}}{\text{TP} + \text{FN}}. \tag{10}$$

$$\text{F1} = \frac{2 \times \text{Precision} \times \text{Recall}}{\text{Precision} + \text{Recall}}. \tag{11}$$

where TP is the number of true positives, FP is the number of false positives, and FN is the number of false negatives.

To compare different models' discriminate ability for the marine aquaculture areas, these accuracy values were calculated for
each class, and the mean F1 values of the MAC and MPC areas were used to assess the performance of the different methods.

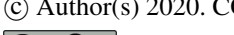



## 4 Results

### 4.1 Spatial distribution of marine aquaculture areas in China

The final classification results are shown in Fig. 5, with the corresponding Google Maps kmz file are published at https://doi.org/10.5281/zenodo.3881612 (Fu et al., 2020). The prediction results of typical areas (Fig. 5a–d) demonstrate the applicability and robustness of the HCHNet method to different marine aquaculture areas (i.e., MPC and MAC) over different study sites (i.e., from Liaoning to Guangxi provinces).

According to the classification results, the total area of marine aquaculture in China is approximately 1103.67 km$^2$. As can be seen from Fig. 6a, marine aquaculture is mainly distributed in the coastal areas of Fujian, Shandong, Liaoning, and Jiangsu provinces. Fujian and Shandong provinces have the largest areas of over 300 and 450 km$^2$, respectively. Furthermore, nearly 100 km$^2$ of marine aquaculture areas are found in Liaoning and Jiangsu provinces, respectively. Fig. 6b shows that over 85% of the marine aquaculture areas in these four coastal areas are MPC. Fig. 6b also shows that the provinces in North China, such as Liaoning and Shandong, tend to have more MPC areas, with the provinces in South China having more MAC areas.

Fig. 6c shows that most of the marine aquaculture areas in China are MPC areas, with an area of over 950 km$^2$, six times larger than the MAC area. Guangxi and Fujian provinces have the largest areas of MAC, which account for more than 70% of the total MAC areas in China (Fig. 6d). The largest areas of MPC are found in Fujian and Shandong provinces, accounting for more than 70% of the total MPC areas in China.

### 4.2 Accuracy assessment of the marine aquaculture area map in China

To quantitatively evaluate the classification performance of our proposed HCHNet method, we used the random stratified sampling method to perform the evaluation. A total of 4000 reference pixels were randomly selected, with 1000 pixels from the classification results for each classes. We then obtained the ground truth of each point by visual interpretation based on HSR images from Google Earth point-by-point.

As shown in Table 2, the error matrix shows that the overall accuracy was 95.83%, and the kappa coefficient was 0.94. The land and sea areas get the highest classification accuracy with both of the PA and UA values greater than 91%. The MPC areas have relatively high PA and UA values of approximately 95%. Most of the misclassifications of MPC areas are related to the sea area. This is because the MPC areas are submerged in a complex sea environment, which can easily be affected by waves, seafloor topography, shadows of clouds, etc. The MAC areas have a relatively lower UA value of 89.1%, which may be caused by the relatively high complexity and small numbers of training patches of MAC areas.

After that, we employed the bootstrapping (Efron and Tibshirani, 1997), which is suitable for estimating classification accuracy (Duan et al., 2020; Lyons et al., 2018), to estimate the uncertainty level. We bootstrapped the overall accuracy from 4000 independent reference points. The bootstrapping was performed with 1000 iterations, and the mean of the distribution used for the evaluation and the confidence intervals was set as 95% quantile. Eventually, we obtained the overall accuracy of





95.8% (95.2%-96.4, 95% confidence interval). Meanwhile, we derived that the marine aquaculture area in China is 1103.67 km$^2$ (1096.8 km$^2$-1110.6 km$^2$, 95% confidence interval).

To further assess the validity of our proposed HCHNet method, we also evaluated the area accuracy (percentage of the over lapping area) based on the test dataset, including 120 randomly selected patches with a size of 256 $\times$ 256, which accounts for 20 % of the labeled samples. As shown in Table 2, the land and sea areas have the best classification accuracy with the area accuracy values greater than 93%. Meanwhile, the MPC and MAC also have a relatively high area accuracy values of 81.8% and 72.5%, respectively.

**4.3 Comparison with the state-of-the-art methods**

To assess the quality of the proposed HCHNet method, the performance was compared with the results from other state-of-the-art FCN-based methods, using the same test dataset. As can be seen from the prediction results in Fig. 7, FCN-32s and Deeplab V2 are unable to reliably identify marine aquaculture, especially the small and isolated marine aquaculture areas (Fig. 7b,d,e) with much coarser predicted results than other approaches. HCHNet identified more MPC and MAC areas than U-Net and HCNet, (Fig. 7a,d,e) and also identified detailed information, even with the narrow channels among neighboring MPC areas (Fig. 7b,e).

To provide a quantitative comparison, several commonly-used accuracy metrics were calculated from the test dataset for MAC and MPC areas. Table 3 shows that the FCN-32s and Deeplab V2 achieved similar accuracy values, with mean F1 values less than 40%. The U-Net and HCNet achieved a similar classification performance, with the mean F1 values of approximately 70%. Compared with these state-of-the-art methods, our proposed HCHNet approach obtained the best classification performance, with the mean F1 value of 76.3%.

**5 Discussion**

**5.1 Date and algorithms for the mapping of marine aquaculture areas in China**

This study developed a new algorithm to separate two typical marine aquaculture types based on the most advanced FCN-based models. The HCHNet was applied to medium spatial resolution images from China's GF-1 WFV sensors to map the marine aquaculture areas in China. The input data and the algorithm used in our study were different from current state-of-the-art methods in many ways.

First, China's GF-1 WFV sensors provide a larger number of valid image scenes that are suitable for a wide range of analyses of marine aquaculture areas with high temporal resolution. MPC areas are only visible in several specific months due to phenological development stages. However, it is difficult to capture appropriate images that clearly represent the marine culture areas in these months from other similar satellites, such as Landsat, which are frequently influenced by clouds or waves. The high temporal resolution of the GF-1 WFV data (repeats each 2 days) means that it is possible to observe



marine aquaculture areas with much greater frequencies than data from other sources. Additionally, the relatively wide swath of the data makes them highly suitable for such large-scale mapping in China. In addition, it is possible to directly obtain images with 16 m spatial resolution without any additional computations, such as pan-sharpening operations, making the GF-1 WFV data a reliable data source for large-scale marine aquaculture area observation and monitoring.

Second, the proposed FCN-based HCHNet method improves the classification accuracy and efficiency. Much previous research has used OBIA approaches (Fu et al., 2019a; Wang et al., 2017) and other FCN-based methods (Fu et al., 2019c; Shi et al., 2018). The accuracy of the OBIA method depends on segmentation, which does not have universal methods for evaluating the selection of appropriate segmentation parameters (Blaschke, 2010). It also takes a large amount of time to undertake the segmentation process and to design effective features or rules for hard classifications (Zheng et al., 2017), making such approaches more difficult to implement operationally for national-scale studies. The HCHNet method achieved the best classification performance for two reasons: (1) all of the pooling operations were removed to improve the identification performance of smaller foreground objects; (2) the hierarchical structure was used to enlarge the receptive field in order to capture more contextual information.

Third, masking out coastal land areas that do not intersect with marine aquaculture areas was undertaken using publicly available data and provided a simple and straightforward methodological refinement to constrain the marine aquaculture mapping. This was important because of the scale of the classification over large coastal areas in China, which contain various land covers outside of the aims of this study. Previous studies have used a threshold value (Zheng et al., 2017) to mask out these land areas, but in this study, this was done directly.

**5.2 Uncertainty and limitations of the marine aquaculture map in China**

Accurate mapping of marine aquaculture areas at a regional scale is challenging. There are several potential uncertainties in our methods for mapping marine aquaculture areas. First, because of the medium spatial resolution imagery and the relatively small size of MAC area (Fig. 8a), it is difficult to accurately identify the boundaries of small and isolated MAC areas (generally less than 10 pixels). Overestimation of MAC may occur, where the sea waters among several MAC areas are misclassified as MAC. The HCHNet failed to detect the small MPC areas (Fig. 8b) and harvested MPC areas (Fig. 8c), causing an issue of underestimation. As shown in Fig. 8d, some vegetation that is submerged or close to the sea waters may be misclassified as MPC areas, since these pixels share similar spectrum and shape features.

There are also some limitations of the proposed HCHNet approach. First, the training process requires a large number of high-quality ground truth labels, which may require much manual work and professional interpretation experience. Therefore, further research on accelerating the training or inference process through weak supervision (Lin et al., 2016; Pathak et al., 2015) or a series of model compression methods (Li et al., 2017; Yim et al., 2017; X. Zhang et al., 2016) will be undertaken to enhance the applicability of the approach. Second, our proposed methods can only be used for the monitoring of marine aquaculture areas in surface water; it is unable to detect the submerged cages in some places (such as coastal area of Shandong province in northeastern China).



## 6 Data availability

The map of marine aquaculture in China's coastal zone at 16 m spatial resolution have been published in the Google Maps kmz File Format with georeferencing information at https://doi.org/10.5281/zenodo.3881612 (Fu et al., 2020).

## 7 Conclusions

Marine aquaculture areas and the coastal environment they rely on are of significant ecological and socioeconomic value. Accurate and effective mapping approaches are imperative for the monitoring, planning, and sustainable development of marine and coastal resources across local, regional, and global scales. The increasing public availability of remote sensing data, ancillary data, and advanced computer vision algorithms, together provided an effective route for identifying marine aquaculture areas at a national scale. By using the powerful and inherent self-learning mechanism of deep learning, a new algorithm was carefully designed based on the FCN structure and applied to the GF-1 WFV data. The application of this algorithm produced a marine aquaculture area map of China with an overall classification accuracy >95% (95.2%-96.4, 95% confidence interval). The total area of China's marine aquaculture areas was estimated to be approximately 1100 km$^2$ (1096.8 km$^2$-1110.6 km$^2$, 95% confidence interval), of which more than 85% is MPC areas. Most of the marine aquaculture areas are distributed along the coastal areas of Fujian, Shandong, Liaoning, and Jiangsu provinces. Guangxi and Fujian provinces have the largest areas of MAC, and Fujian and Shandong have the largest areas of MPC. The algorithm could be implemented at other regional and global scales with the collection of sufficient samples and the careful investigation of marine aquaculture phenology in these areas.

## Author Contributions

Funding acquisition, J.D., W.Y and A.C.; methodology, Y.F.; supervision, J.D. and K.W.; writing—original draft, Y.F.; writing—review & editing, H.W., A.C., S.Y., Y.L., and W.W.

## Acknowledgements

This research was funded by the National Natural Science Foundation of China (71974171), Ministry of Science and Technology of China (2016YFC0503404), Natural Science Foundation of Zhejiang Province (LY18G030006), Science and Technology Department of Zhejiang Province (2018F10016), and the Natural Environment Research Council (NE/E523213/1).



**Competing interests**

The authors declare that they have no conflict of interest.

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

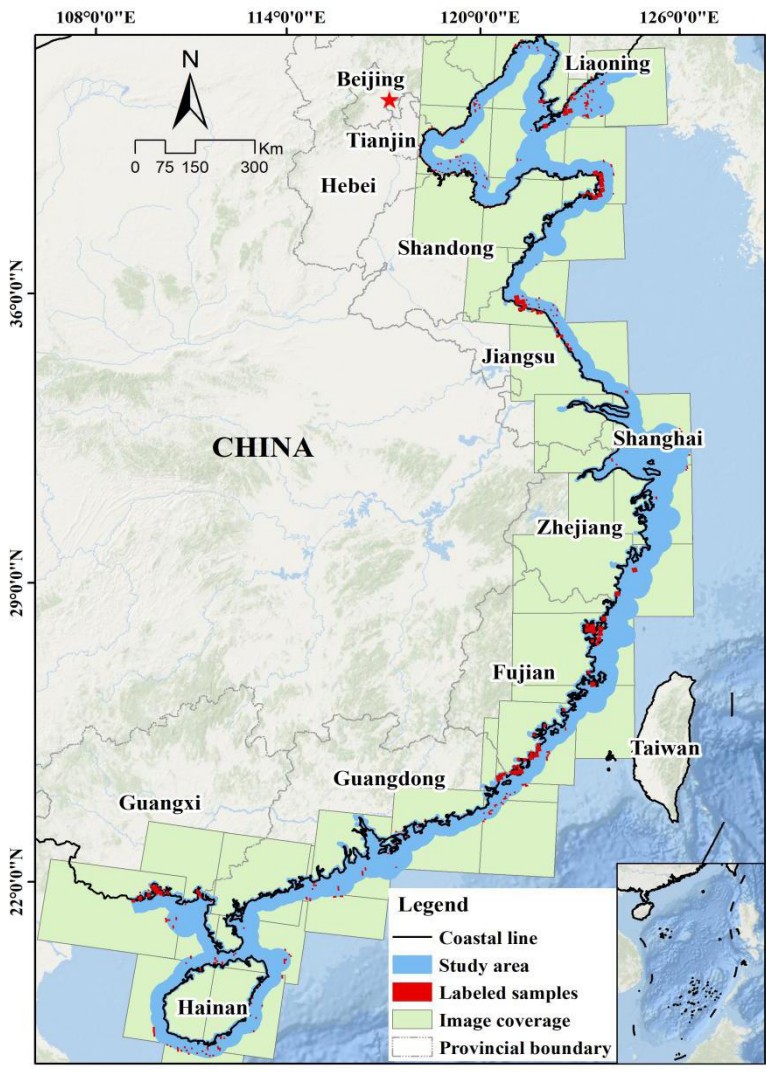

**Figure 1: Location of the study area, the spatial distribution of labeled samples, and acquired GF-1 wide-field-of-view (WFV) image swaths in China.**

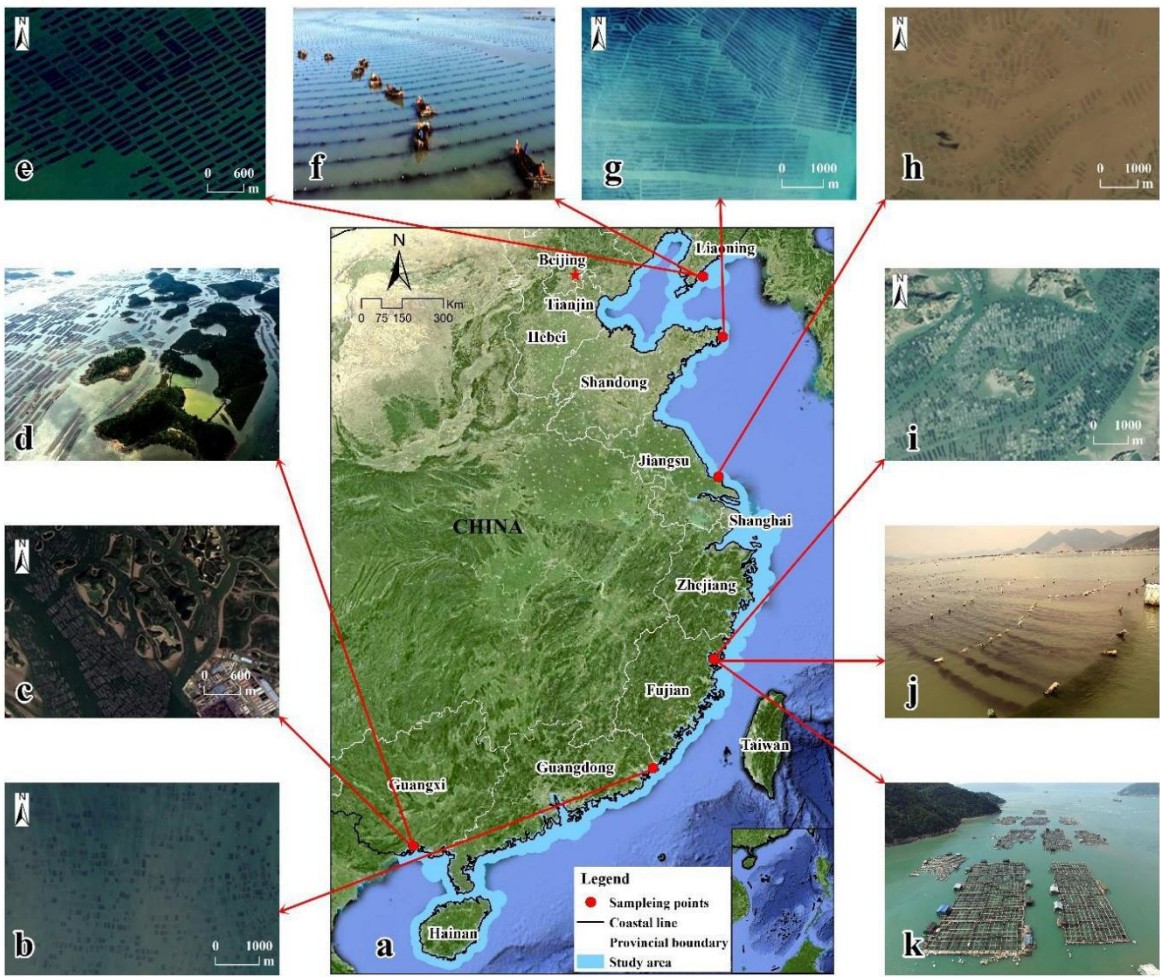

**Figure 2: Location of the Sampling points (a). And image examples of typical marine aquaculture areas on ground or from high spatial resolution (HSR) images: (b) (e) (g) (h) (i) Marine plant culture (MPC) areas from HSR images; (c) (i) Marine animal culture (MAC) areas from HSR images; (d) (k) photos of MAC areas on ground; (f) (j) photos of MPC areas on ground.**



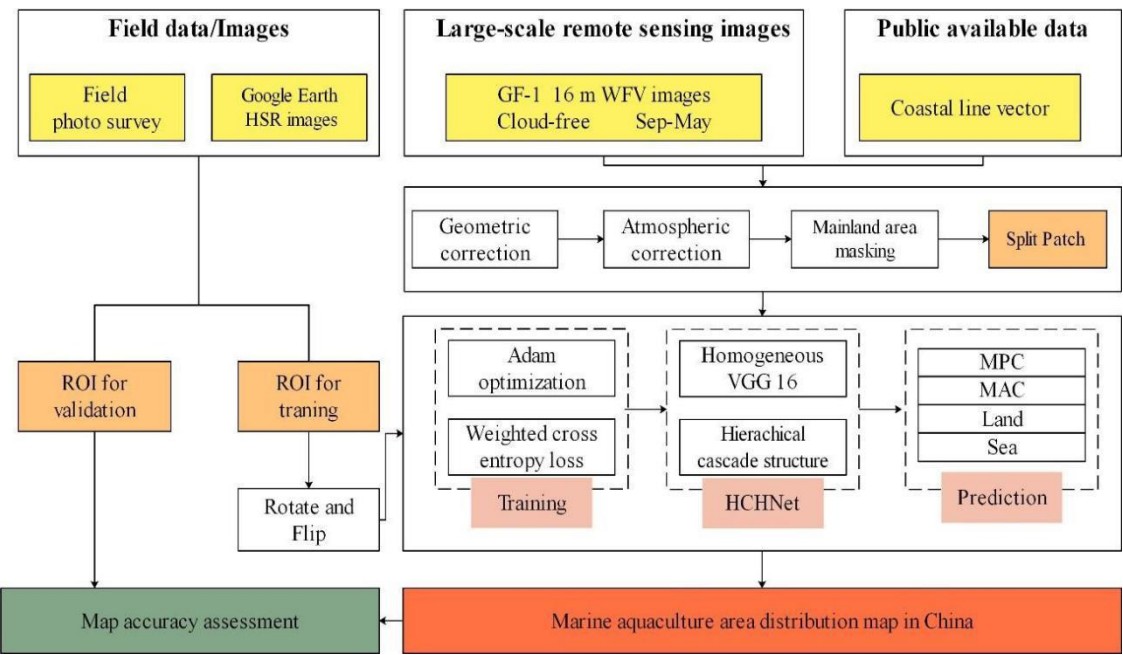

**Figure 3: Schematic flowchart of the marine aquaculture mapping (ROI: region of interest. HCHNet: hierarchical cascade homogeneous neural network).**


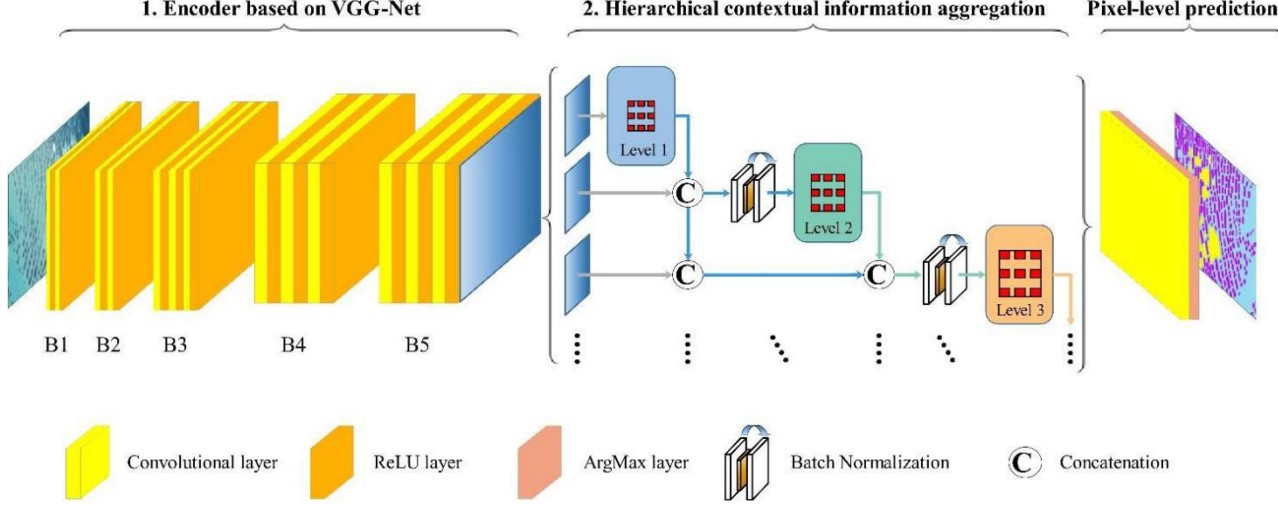

**Figure 4: Overview of the proposed HCHNet approach.**





**Figure 5: Spatial distribution of China's marine aquaculture and zoom views of imagery and prediction results for typical**
**areas of (a), (b), (c), and (d).**

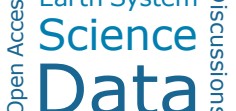



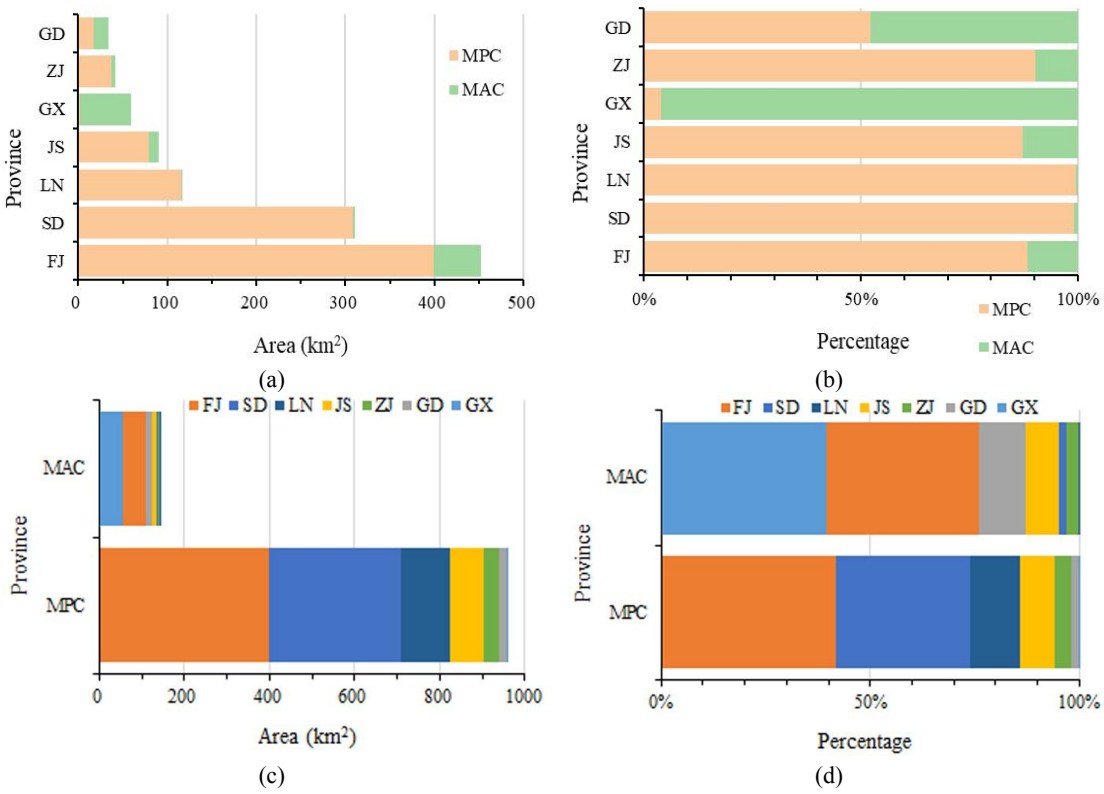

**Figure 6: The amount (area) and percentage of different types of marine aquaculture distribution in the coastal provinces of China.LN, SD, JS, ZJ, FJ, GX, and GD indicate Liaoning, Shandong, Jiangsu, Zhejiang, Fujian, Guangxi, and Guangdong provinces, respectively.**





**Figure 7: The classification results of MPC and MAC areas comparing the proposed HCHNet method with other approaches.**
**The black outlined areas indicate where HCHNet obtains better results. The red, yellow, blue, and green areas in the**
**classification maps represent the MPC, MAC, sea, and land areas, respectively.**

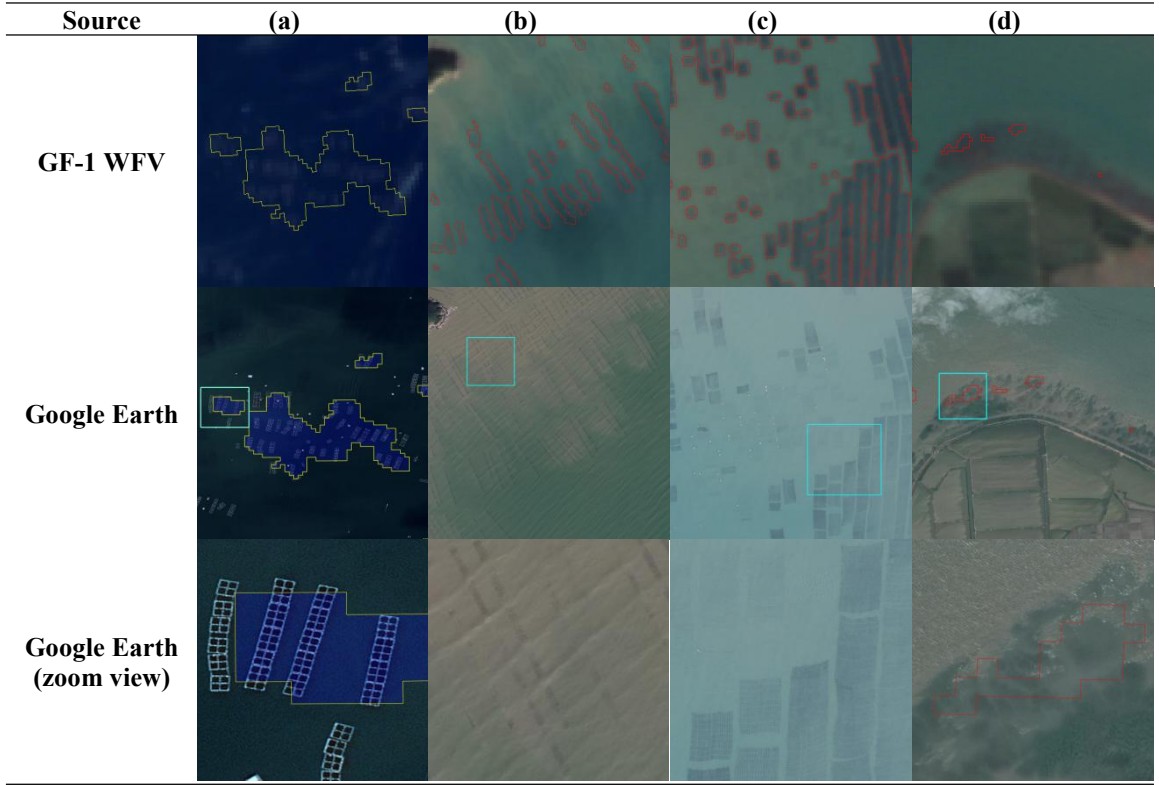

**Figure 8: Illustration of potential sources of error in the HCHNet algorithm: the boundaries of relatively small and separate MAC (a) or MPA (b) areas are difficult to accurately identify; (c) harvested MPC areas are also difficult to detect due to shallow waters and the disappeared dark tone; (d) vegetation located close to the water bodies may be misclassified as MPC areas. The first row highlights typical misclassified areas from GF-1 WFV data. The second row shows high spatial resolution images from Google Earth (© Google Maps). The third row is a zoom view of the Google Earth images in the second row (© Google Maps). The red and yellow areas indicate the classification results of MPC and MAC areas, respectively.**



Table 1: Detailed configuration of the encoder in the proposed HCHNet method. (*l, k×k×n, s*) of configurations means there are *l* convolution layers with n convolution kernels and their size is *k×k* and stride is *s*. (*h, w, c*) of the output size means the output image or feature maps have a height of *h*, a width of *w*, and a channel of *c*.

| Layer name | Layer type | Configurations | Output size |
|---|---|---|---|
| Input | Image data | - | 256 × 256 × 4 |
| B1 | Convolution, ReLU | 2, 3 × 3 × 32, 1 | 256 × 256 × 32 |
| B2 | Convolution, ReLU | 2, 3 × 3 × 64, 1 | 256 × 256 × 64 |
| B3 | Convolution, ReLU | 3, 3 × 3 × 128, 1 | 256 × 256 × 128 |
| B4 | Convolution, ReLU | 3, 3 × 3 × 256, 1 | 256 × 256 × 256 |
| B5 | Convolution, ReLU | 3, 3 × 3 × 256, 1 | 256 × 256 × 256 |

Table 2: Accuracy assessment of the classification results in China based on visual interpretation (pixels). (PA: producer accuracy. UA: user accuracy)

| Predicted class | Ground truth | | | | | |
|---|---|---|---|---|---|---|
| | Sea | Land | MPC | MAC | Sum | UA |
| Sea | 981 | 12 | 5 | 2 | 1000 | 98.10% |
| Land | 5 | 987 | 3 | 5 | 1000 | 98.70% |
| MPC | 20 | 3 | 974 | 3 | 1000 | 97.40% |
| MAC | 31 | 76 | 2 | 891 | 1000 | 89.10% |
| Sum | 1037 | 1078 | 984 | 901 | | |
| PA | 94.60% | 91.56% | 98.98% | 98.89% | | |
| Overall accuracy | 95.83% | | | | | |
| Kappa coefficient | 0.94 | | | | | |

Table 3. Quantitative comparison of MPC and MAC areas between our method and other methods, where the best accuracy values are in bold (%).

| Methods | MPC | | | MAC | | | Avg. |
|---|---|---|---|---|---|---|---|
| | Precision (%) | Recall (%) | F1 (%) | Precision (%) | Recall (%) | F1 (%) | mean F1 (%) |
| FCN-32s | 66.1 | 54.2 | 59.6 | 63.9 | 8.60 | 15.2 | 37.4 |
| DeeplabV2 | 70.4 | 40.9 | 51.7 | 53.7 | 5.71 | 10.3 | 31.0 |
| U-Net | 83.8 | 77.6 | 80.6 | 81.8 | 53.1 | 64.4 | 72.5 |
| HCNet | 77.3 | 79.0 | 78.1 | 74.0 | 40.0 | 51.9 | 65.0 |
| Ours-HCHNet | 82.2 | 81.8 | **82.0** | 68.7 | 72.5 | **70.6** | **76.3** |
