# Peer review of "A new satellite-derived dataset for marine aquaculture areas in the China's coastal region"

_Earth System Science Data, 2020_

## Referee Comment (RC1) · Anonymous Referee #1 · 24 Jan 2021

This study aimed to produce an accurate national-scale marine aquaculture map at a spatial resolution of 16 m. I can see authors put lots of effort into data processing, experiments and comparisons. However, I do have some significant concerns.

First of all, in the title and the abstract, it claims that this study aims to propose a new dataset of the classification results covering the entire coastline of China based on a new data source that using GF-1 and similar at 16 m spatial resolution. Yet in the text of the manuscript, most parts focused on the utilization of convolution neural networks and the classification method that is advanced than the previous methods.

If the innovation part is the CNN based algorithm, a more challenging experiment (or an artificial toy data manipulation) should be select to illustrate and emphasize the classification performance results, theoretically and empirically.

Currently, it is not significant to see how the algorithm outperforms previous methods from Fg 7. The classification case selections are straightforward (the classes are obvious in the original image). The traditional pixel-based or object-based classification results might obtain similar results if set the parameters correctly.

If the new method is advanced in all cases on the coast of China, the method is appliable to the entire coast. If there are previous literature already manifested this point, to cite this in the introduction should be adequate.

Actually, The longer coast involves lots of the local variations that could affect the image processing results. It is hardly foreseen that the new method could outperform all other methods in the coastal regions across such larger latitudes. It might be better to focus on a few specific applications for the method's better performance.

Other comments,

I think no need to make data availability as a single section. This can go to data, results, or appendix.

L105, consider to specify the atmosphere profiles and other in situ data used in FLAASH atmospheric correction.

---

## Referee Comment (RC2) · Anonymous Referee #2 · 26 Jan 2021

Review of ESSD-2020-122 This manuscript presents a study where a deep convolutional neural network was designed and built to purposefully map marine agriculture at 16 meter resolution. The manuscript is well written and presents clear objective. My comments are:

First, the authors should sharpen the subject of this study. Currently the focus of this study is on comparison of the new algorithm and other methods, while the feature or content of GF-1 data, as a new data source, is better to provide more info. Another question is about the fine-tuning of various methods in the comparison. To make a fair comparison, are all the methods fine-tuned to their optimal status for the classification. Currently, this is not very clear. Also, the advantage of the proposed algorithm over the U-Net and HCNet is not very clear (also see minor comment # 5). This point deserves more clarification and discussion. Minor comments: 1. In the caption of Figure 7, the MPC area should be described as purple, instead of red. 2. Line 94: 'environment' should be plural, i.e. 'environments'. 3. Line 194: 'To perform the accuracy assessment' is a very general purpose. If not mistaken, the stratified sampling is done to ensure representativeness of each class in the whole sample population. I recommend state it more specifically. 4. Line 235: should be 'performed for 1000 iterations'. 5. Line 248: 'HCHNet identified more MPC and MAC areas than U-Net and HCNet, (Fig. 7a,d,e).' This statement is not visually obvious, especially in the d and e cases between UNet and HCNet. I suggest the authors rephrase or be more specific when making this comparison. 6. Line 276: should be '. . . more difficult to be implemented . . .'

---

## Referee Comment (RC3) · Anonymous Referee #3 · 28 Jan 2021

This study proposed and applied a deep learning based method to the medium spatial resolution images at national scale. In the produced dataset, types and locations of marine aquaculture are described in a detailed way, which fill in the gap of this data in China. The topic of the study is interesting and fits the scope of the journal. However, main innovations of this study should be stated more clearly. And there are still some problems that need more explanation. Here with concerns need to be addressed: Question & Comment 1: Compared with the existing method, such as the other deep learning based methods, what are the differences or improvements in the structure of the proposed methods? It is the core value of this study. Thus, main innovations and contributions should be summarized more clear. Question & Comment 2: Have you considered the similar medium-spatial resolution images that may have more spectrum

information, such as Landsat or Sentinel? I suggest more description about the data choice. Question & Comment 3: Line 31: The production of marine aquaculture in 2017 should be replaced with the most recent statistical data. Question & Comment 4: Line 190: There are many other state-of-the-art models in the computer vision fields, why choose the above models for comparison? More reasons should be given for such selection. Question & Comment 5: Line 480: I suggest that more obvious differences should be point out in this figure. Question & Comment 6: Line 495: As illustrated in the line 280, this study used the publicly available data to mask out coastal land areas that do not intersect with marine aquaculture areas. Why the table still shows accuracy values of the land area?

---

## Author Comment (AC1) · 1 Mar 2021

**Response to Reviewers' Comments**

**We would like to thank the reviewer for all the constructive comments, which have improved the manuscript significantly. A detailed response to all comments can be found below, where the comments are in regular font and our point-to-point responses are in bold font. Line numbers correspond to the revised manuscript.**

Comments:

- This study aimed to produce an accurate national-scale marine aquaculture map at a spatial resolution of 16 m. I can see authors put lots of effort into data processing, experiments and comparisons. However, I do have some significant concerns.
  **Response: Thanks for pointing out our main contributions and the suggestions which for sure significantly improved the manuscript.**
- First of all, in the title and the abstract, it claims that this study aims to propose a new dataset of the classification results covering the entire coastline of China based on a new data source that using GF-1 and similar at 16 m spatial resolution. Yet in the text of the manuscript, most parts focused on the utilization of convolution neural networks and the classification method that is advanced than the previous methods.
  **Response: Apparently, the main contributions were not clearly explained in the original paper. Our main contributions include both the new dataset and classification method.To make this point clear, we added several paragraphs to the introduction section (bellow and in the revised paper).**
  **Line 68-74:**
  **'To overcome these limitations, we proposed a novel framework for the large-scale marine aquaculture mapping. The main contributions of our study can be summarized as follows:**
  **(1) We present a unified CNN-based framework for national-scale marine aquaculture extraction.**
  **(2) A hierarchical cascade homogeneous neural network (HCHNet) model is proposed to learn discriminative and robust features.**
  **(3) We provide the first detailed national-scale marine aquaculture map with a spatial resolution of 16 m. '**
- If the innovation part is the CNN based algorithm, a more challenging experiment (or an artificial toy data manipulation) should be select to illustrate and emphasize the classification performance results, theoretically and empirically.
  **Response: Revised as suggested. We conduct more experiments based on the artificial toy data, which is called test dataset in the study, to better illustrate and emphasize the classification performance results.**
  **Line 207-210:**
  **'Meanwhile, we also conducted the area accuracy assessment (the**

**percentage of overlapping areas in the ground truth) based on more than 120 randomly selected 256×256 patches, which accounts for nearly 20% of the total samples.'**

**The results are shown in the new Table 3:**

**Table 3. Area accuracy assessment of different classes based on randomly selected patches.**

| Class | Sea area | Land area | RCA | CCA |
|---|---|---|---|---|
| **Area accuracy:** | 93.6% | 98.4% | 81.8% | 72.5% |

**Besides, we also discussed more about the reason why our proposed methods can obtain higher accuracy values.**

**Line 289-294:**

**'The proposed HCHNet achieved the best classification performance for three reasons: (1) all of the pooling operations were removed to avoid the shrinking of features, which helps improve the identification of smaller foreground objects; (2) the hierarchical structure was used to enlarge the receptive field to capture more contextual information, which is helpful for reducing the influence of local variance; (3) the weighted loss function was employed to solve the classes imbalance problem.'**

● Currently, it is not significant to see how the algorithm outperforms previous methods from Fig.7.

**Response: Revised as suggested. We revised the Fig.7 to show clearly the advantages of the proposed algorithm. As shown in the new version of Fig.7, the black solid outlined areas clearly indicate where our methods obtained better results.**

[Figure]

**Figure 7: The classification results of MPC and MAC areas comparing the proposed HCHNet method with other approaches. The black solid outlined areas indicate where HCHNet obtains better results. The dotted line shows same locations in other images.The purple, yellow, blue, and green areas in the classification maps represent the MPC, MAC, sea, and land areas, respectively.**

- The classification case selections are straightforward (the classes are obvious in the original image).

  **Response: To obtain an accurate classification result, we only employed cloud-free images in this study. Thus, the original images used in this study, which also include the selected cases, are straightforward.**

- The traditional pixel-based or object-based classification results might obtain

similar results if set the parameters correctly.

**Response: Compared with the traditional pixel-based or object-based classification methods, CNN-based method can also achieve good balance between higher accuracy values and generality(https://doi.org/10.1109/MGR S.2016.2540798). Most importantly, such methods do not need to design complex features or set parameters for local variances as traditional methods. Thus, the CNN-based methods are more suitable for the large scale classification.**

- If the new method is advanced in all cases on the coast of China, the method is appliable to the entire coast. If there are previous literature already manifested this point, to cite this in the introduction should be adequate.

  **Response: To our best knowledge, we provided the first detailed marine aquaculture map at the national scale using CNN-based methods. Technically, such CNN-based method can be easily applied to larger scales (https://doi.org/10.3390/rs12010002;https://doi.org/10.1109/TGRS.2019.2904 868 ). However, considering the local differences among countries, we will improve our method and provide new dataset across the entire coast in our future works.**

- Actually, The longer coast involves lots of the local variations that could affect the image processing results. It is hardly foreseen that the new method could outperform all other methods in the coastal regions across such larger latitudes. It might be better to focus on a few specific applications for the method's better performance.

  **Response: To overcome such difficulties, we made several important steps in the classification process: First, we choose to built our models based on the CNN architecture, which have a good balance between accuracy and robustness; Second, we trained our model with randomly selected samples across the coastal regions. Besides, our model are designed to have a larger reception field, which is specially helpful for a wide range of local variations; Third, we used the atmosphere correction and precious coastal line to reduce the influence of local variations. Therefore, our proposed methods can work well in the coastal regions across such larger latitudes.**

- I think no need to make data availability as a single section. This can go to data, results, or appendix

  **Response: As requested in the 'Submission-Get ready' section in the official website (https://www.earth-system-science-data.net/submission.html), we need to include the DOI of the proposed dataset in both the abstract and the data availability section. Therefore, we may need to remain this section as the published articles.**

- L105, consider to specify the atmosphere profiles and other in situ data used in FLAASH atmospheric correction.

  **Response: Revised as suggested.**
      **Line 117-118:**
      **'The "Maritime" model was set as Aerosol Model. And all the other**

parameters can be automatically set by using the extension tools
( https://github.com/yyong-fu/ENVI_FLAASH_EasyToUse) '

---

## Author Comment (AC2) · 1 Mar 2021

**Response to Reviewers' Comments**

**We would like to thank the reviewer for all the constructive comments, which have improved the manuscript significantly. A detailed response to all comments can be found below, where the comments are in regular font and our point-to-point responses are in bold font. Line numbers correspond to the revised manuscript.**

Comments:

- Review of ESSD-2020-122 This manuscript presents a study where a deep convolutional neural network was designed and built to purposefully map marine agriculture at 16 meter resolution. The manuscript is well written and presents clear objective. My comments are:
  **Response: Thanks for the positive comments and all the suggestions, which significantly helps we improve our manuscript.**
- First, the authors should sharpen the subject of this study. Currently the focus of this study is on comparison of the new algorithm and other methods, while the feature or content of GF-1 data, as a new data source, is better to provide more info.
  **Response: Revised as suggested. We added more introductions of the GF-1 to the data section (bellow and in the revised paper).**
    **Line 104-110:**
    **'The GF-1 satellite, which is the first satellite of the China high-resolution earth observation satellite program, was launched by the China Aerospace Science and Technology Corporation in April 2013. This satellite carries four integrated WFV sensors, providing multi-spectrum data with a two-day revisit cycle and a swath width of 800 km when the four sensors are combined. Each WFV sensor has four multi-spectral bands at 16 m spatial resolution: B1 (450–520 nm, blue), B2 (520–590 nm, green), B3 (630–690 nm, red), and B4 (770–890 nm, near infrared).'**
- Another question is about the fine-tuning of various methods in the comparison. To make a fair comparison, are all the methods fine-tuned to their optimal status for the classification.Currently, this is not very clear.
  **Response: Revised as suggested. All the compared methods are trained from scratch. We made this point clear in the comparison section.**
    **Line 199-201:**
    **'In the training phase, all of the above models, including the proposed HCHNet, were trained from scratch using the same patches and experimental settings as in the HCHNet method. '**
- Also, the advantage of the proposed algorithm over the U-Net and HCNet is not very clear (also see minor comment # 5). This point deserves more clarification and discussion.

**Response: Revised as suggested.We firstly revised the Fig.7 to show the advantages of the proposed algorithm. As in the last column of the Fig.7 , UNet or HCNet tend to recognize the MAC as others, leading to lower recall values.**

[Figure]

**Figure 7: The classification results of MPC and MAC areas comparing the proposed HCHNet method with other approaches. The black solid outlined areas indicate where HCHNet obtains better results. The dotted line shows same locations in other images.The purple, yellow, blue, and green areas in the classification maps represent the MPC, MAC, sea, and land areas, respectively.**

**In addition, we also make this point clear in the results analysis part.**

**Line 266-268:**

**'Besides, the HCHNet also achieved a good balance between precision and recall values of MAC, identifying more accurate and existing MAC areas. The difference between them is less than 4% for the HCHNet, while the difference values of other methods are more than 28%.'**

**And then, we discussed more to analysis the reasons.**

**Line 289-294:**

**'The HCHNet achieved the best classification performance for three reasons: (1) all of the pooling operations were removed to avoid the shrinking of features, which helps improve the identification of smaller foreground objects; (2) the hierarchical structure was used to enlarge the receptive field to capture more contextual information, which is helpful for reducing the influence of local variance; (3) the weighted loss function was employed to solve the classes imbalance problem.'**

Minor comments:

- 1. In the caption of Figure 7, the MPC area should be described as purple, instead of red.

  **Response: Revised as suggested.**

  **'The purple, yellow, blue, and green areas in the classification maps represent the MPC, MAC, sea, and land areas, respectively.'**

- 2. Line 94: 'environment' should be plural, i.e. 'environments'.

  **Response: Revised as suggested.**

  **'the features of MPC in remotely sensed images are usually influenced by different environments (Fig. 2b, e, g, h , j), making it difficult for classification.'**

- 3. Line 194: 'To perform the accuracy assessment' is a very general purpose. If not mistaken, the stratified sampling is done to ensure representativeness of each class in the whole sample population. I recommend state it more specifically.

  **Response: Revised as suggested.**

  **'To ensure representativeness of each class in the whole sample population for accuracy assessment, we followed the widely used random stratified sampling method (Padilla et al., 2014; Ramezan et al., 2019) to generated 4000 randomly selected points in the coastal zone.'**

- 4. Line 235: should be 'performed for 1000 iterations'.

  **Response: Revised as suggested.**

  **'The bootstrapping was performed for 1000 iterations, and the mean of the distribution used for the evaluation and the confidence intervals was set as 95% quantile.'**

- 5. Line 248: 'HCHNet identified more MPC and MAC areas than U-Net and HCNet,(Fig. 7a,d,e).' This statement is not visually obvious, especially in the d and e cases between UNet and HCNet. I suggest the authors rephrase or be more

specific when making this comparison.

**Response: Revised as suggested.We revised the Fig.7 to show the advantages of the proposed algorithm. The revised Fig.7 and statements can be found in previous comments.**

- 6. Line 276: should be '. . . more difficult to be implemented

  **Response: Revised as suggested.**

  **'making such approaches more difficult to be implemented operationally for national-scale studies.'**

---

## Author Comment (AC3) · 1 Mar 2021

**Response to Reviewers' Comments**

We would like to thank the reviewer for all the constructive comments, which have improved the manuscript significantly. A detailed response to all comments can be found below, where the comments are in regular font and our point-to-point responses are in bold font. Line numbers correspond to the revised manuscript.

Comments:

● This study proposed and applied a deep learning based method to the medium spatial resolution images at national scale. In the produced dataset, types and locations of marine aquaculture are described in a detailed way, which fill in the gap of this data in China. The topic of the study is interesting and fits the scope of the journal. However, main innovations of this study should be stated more clearly. And there are still some problems that need more explanation. Here with concerns need to be addressed:

**Response: Thanks for pointing out the contributions of our study and all the suggestions that improves our manuscript significantly.**

● Question & Comment 1: Compared with the existing method, such as the other deep learning based methods, what are the differences or improvements in the structure of the proposed methods? It is the core value of this study. Thus, main innovations and contributions should be summarized more clear.

**Response: Revised as suggested. Our main contributions are summarized and added to the introduction section (bellow and in the revised paper).**

> **Line 68-74:**
> **'To overcome these limitations, we proposed a novel framework for the large-scale marine aquaculture mapping. The main contributions of our study can be summarized as follows:**
> **(1) We present a unified CNN-based framework for national-scale marine aquaculture extraction.**
> **(2) A hierarchical cascade homogeneous neural network (HCHNet) model is proposed to learn discriminative and robust features.**
> **(3) We provide the first detailed national-scale marine aquaculture map with a spatial resolution of 16 m. '**

● Question & Comment 2: Have you considered the similar medium-spatial resolution images that may have more spectrum information, such as Landsat or Sentinel? I suggest more description about the data.

**Response: Revised as suggested. Compared with the similar medium-spatial resolution images, we selected the GF-1 WFV images as our data sources for the higher temporal resolution and relatively wider swath. We added several sentences to the data section make this point more clear.**

> **Line 109-111:**
> **'Compared with other frequently used medium resolution satellite**

imagery (e.g. Landsat, Sentinel), the wide coverage ability, high-frequency revisit time, and 16-m spatial resolution of the data significantly improves the capabilities for large-scale marine aquaculture areas observation and monitoring.'

- Question & Comment 3: Line 31: The production of marine aquaculture in 2017 should be replaced with the most recent statistical data

  **Response: Revised as suggested.**

  **Line 30-31:**

  **'The marine aquaculture production in China has increased from 10.6 million tons in 2000 (Bureau of Fisheries of the Ministry of Agriculture, 2001) to 20.7 million tons in 2019 (Bureau of Fisheries of the Ministry of Agriculture, 2020).'**

- Question & Comment 4:Line 190: There are many other state-of-the-art models in the computer vision fields,why choose the above models for comparison? More reasons should be given for such selection.

  **Response: Revised as suggested. We selected these models for their similar encoders and the typical multi-scale structures, which are more suitable for comparison with the proposed hierarchical cascade structure. We added such expressions in the comparing section.**

  **Line198-199:**

  **'The above models are suitable for classification and comparison purposes, because nearly all of these methods are VGG-16 based neural networks and employed typical structures for multi-scale information extraction.'**

- Question & Comment 5: Line 480: I suggest that more obvious differences should be point out in this figure.

  **Response: Revised as suggested. We revised the Fig.7 to show the advantages of the proposed algorithm. As shown in the new version of Fig.7, the black solid outlined areas clearly indicate where our methods obtained better results.**

[Figure]

**Figure 7: The classification results of MPC and MAC areas comparing the proposed HCHNet method with other approaches. The black solid outlined areas indicate where HCHNet obtains better results. The dotted line shows same locations in other images.The red, yellow, blue, and green areas in the classification maps represent the MPC, MAC, sea, and land areas, respectively.**

- Question & Comment 6: Line 495: As illustrated in the line 280, this study used the publicly available data to mask out coastal land areas that do not intersect with marine aquaculture areas. Why the table still shows accuracy values of the land area?

  **Response: In our study, we used the publicly available data to mask out**

coastal land areas. However, some of the small land areas, such as island or seashore, are not include in the coastal land areas. Meanwhile, some of the land areas in the public data may be changed by the land reclamation project.Thus, the table still shows accuracy values of the land area.

---

## Author Response (AR1)

Our point-to-point response to the reviewers have been presented in the public responses to their comments. In particular, we have:

1. Summarized the main contributions, in response to the issue brought up by Reviewer 1,3, see line 68-74 in the revised manuscript.

2. Added a new Table 3 and corresponding analysis in line 289-294 to emphasize the classification performance, in response to a suggestion by Reviewer 1.

3. Revised the Fig.7 to show clearly the advantages of the proposed algorithm, in response to a suggestion by Reviewer 1,2.

4. Added more descriptions about the GF-1 data in response to the suggestion by Reviewer 2,3, see line 104-110.

5. Addressed a few small clarifications raised by Reviewer 1,3.

6. Addressed a number of minor points suggested by Reviewer 2.

7. Corrected some typos, and added a few explanations suggested by Reviewer 2.

[revised manuscript text omitted]